# OpenReview forum: "Fixing the Broken Compass: Diagnosing and Improving Inference-Time Reward Modeling"
_ICLR.cc/2026/Conference — ICLR 2026 Poster_

### Official Review · Reviewer_nfv4 · 2025-10-28

**Soundness:** 3
**Presentation:** 3
**Contribution:** 3
**Rating:** 4
**Confidence:** 4

**Summary:**

This paper first presents observations regarding failure modes of inference time methods that use reward models (RM): (1) RM method perform can produce worse performance on easier questions defined based on whether the model can already likely answer them, (2) RMs' tendency to attribute higher scores to rate, incorrect responses, and (3) excessively high search diversity may lead to worse RM performance.

Based on these observations, the paper proposes a method (CRISP) that (a) generates reasoning paths and cluster them based on answers, (b) aggregates reward signals based on clusters and selects the highest score paths from the top cluster to generate prefixes for the next iteration, (c) terminates the iterations early if only a small number of clusters are left. The paper empirically shows that CRISP outperforms other RM methods on various reasoning benchmarks.

**Strengths:**

- The paper is well-written, with observations on key issues around RM-based inference time methods first presented before outlining the CRISP methodology with components that attempt to address each of these key issues: (1) Early termination addresses poorer performance on easy questions, (2) Cluster aggregation is motivated to mitigate the inverse long-tail issue, and (3) path generation attempts to constrain search diversity.

- The paper addresses an important problem of how to better design performant inference time methods while also reducing token costs.

- The paper presents extensive experimental results, with empirical studies on various failure modes of RMs, comparisons with different RMs, base models and benchmarks.

**Weaknesses:**

- The paper has relatively weak theoretical justification. While the paper attempts to motivate the method's various components loosely by linking them with the observed empirical observations, the design of CRISP remains largely heuristics-based.

- The authors may want to justify some of their experimental design more clearly on why the results are meaningful.
    - For example, for assessing question-difficulty they used the model's own responses to bin the question. As the 'easier' questions were already found to be answerable by the model itself, it is no surprise that applying additional RM methods will not result in better performance.
    - The inverse long-tail point could be better substantiated and made clearer, especially if it is a central claim of the paper. It is not clear whether Fig.5 is sufficient evidence. If the claim is that high rewards of RMs should be distrusted when sample frequency from the base model is low, this seems counter-intuitive since the sample frequency depends on the base model being queried rather than be a property of the RM. Showing that among high RM score samples, frequency of the samples is a good classifier for actual accuracy, could  potentially help, though there should be more thorough analysis.
    - For the illustration on accuracy versus coverage, without normalization it may not be surprising that the count of the number of times 'correct-to-wrong' transitions occur is increasing with N. Directly assessing diversity metrics of sampled responses and performance may be more meaningful too.

- The performance gains presented (e.g. Table 1 and 2) sets to not be that large for many cases (especially given the lack of error bars). The authors may want to discuss some of these results more thoroughly. Table 2 seems to indicate that non-math tasks may have different performance trends, but benchmark method results were not presented for them in Table 1.
   - The performance gains should be contrasted with the additional computational costs. E.g. CRISP takes close to 60% more time compared to BoN for GSM8K but seem to achieve the same performance (0.95 accuracy).

- For cost comparisons, an important factor is whether the proposed method scales well. The authors could consider including results on whether the proposed method scales well over N and number of iterations, compared to other baseline methods.

- The proposed approach combines the benefits of 'self-consistency'-type sampling-based approaches and RM scoring, but there may be issues related to it that the paper could more fully analyze especially given the weak theoretical justifications. For example, given two clusters, (1) with many samples, each with middle-level RM scores, and (2) with fewer (but still not rare) samples each with very high RM scores, the proposed approach may select (1) instead of (2) which intuitively may not be the right approach. The specific CRISP approach may seem sensitive to such thresholding. Exploration of different aggregation strategies within the framework could be useful.

- The paper could benefit from a fuller discussion of related work, to contrast its contributions to existing work.  For example, the authors could be more explicit in discussing among the key failure modes of RM methods, whether there have already been existing work that have investigated similar problems and have proposed methods [1,2]. There could also be comparisons/discussion against works that optimizes prompts/prefixes to improve sampling diversity of reasoning paths, including those that work with RMs [3,4].



[1] Chen et al, When is Tree Search Useful for LLM Planning? It Depends on the Discriminator

[2] Stroebl et al, Inference Scaling fLaws: The Limits of LLM Resampling with Imperfect Verifiers

[3] Hu et al, Dipper: Diversity in Prompts for Producing Large Language Model Ensembles in Reasoning Tasks

[4] Du et al, Improving factuality and Reasoning in Language Models through Multiagent Debate

**Questions:**

Please see questions already embedded in the weaknesses section.

---

> ### Author Response · Authors · 2025-11-14
> **Responses to Reviewer nfv4 (part 1)**
>
> Thanks for your careful and insightful reviews.
>
> **Weakness 1**
>
> > While the paper attempts to motivate the method's various components loosely by linking them with the observed empirical observations, the design of CRISP remains largely heuristics-based.
>
> **Reply 1**
>
> We apologize for the misunderstanding caused by the unclear explanation in this part. In fact, we provide a theoretical analysis of the effectiveness of our method in **Appendix K**, and we refer to it in line 353. Overall, our theoretical analysis shows that incorporating occurrence frequency into the optimization of RM scoring can mitigate the issue identified in Figure 5, where the RM assigns overly high scores to low-frequency negative examples. This property is also central to the design of CRISP. Therefore, the design of our method is not merely heuristic but is informed by both empirical findings and theoretical analysis.
>
>
>
> **Weakness 2**
>
> > The authors may want to justify some of their experimental design more clearly on why the results are meaningful.
>
> **Reply 2**
>
> Thank you for your suggestion, it is crucial for the improvement of our paper.
>
> + For question difficulty:
>
>   + Although the generative model performs well on these questions, this does not necessarily mean that introducing the RM will reduce performance, because **the two evaluations focus on different model types**. Since the generative model can reliably recall the correct answers on simple questions, a sufficiently strong RM should allow BoN to achieve accuracy close to 100 percent on these cases, further improving the performance compared to SC. However, our experimental results show the opposite, which suggests limitations in the RM.
>
>   + In addition, we introduce the difficulty level from the original MATH-500 dataset [1], **which is independent of any specific model**, in order to more objectively compare the performance of different paradigms across varying difficulty levels. The results are shown in the table (Qwen2.5-3B):
>
>     |         | SC       | BoN (ORM) | BoN (PRM) |
>     | ------- | -------- | --------- | --------- |
>     | Level 1 | **0.93** | 0.86      | 0.79      |
>     | Level 2 | **0.80** | 0.72      | 0.71      |
>     | Level 3 | **0.76** | 0.73      | 0.71      |
>     | Level 4 | 0.61     | **0.62**  | 0.60      |
>     | Level 5 | 0.37     | 0.38      | **0.46**  |
>
>     The results show that the question difficulty in our findings is actually independent of the specific model.
>
> + For inverse long-tail point:
>
>   + First, we believe that our experimental results provide sufficient evidence to support the inverse long-tail phenomenon. In Figure 5, we evaluate multiple RMs and observe the same distribution pattern across all of them, and in Figure 23, we further show that this phenomenon is independent of the sampling strategy. If you think additional experiments are needed in this direction, we would be happy to follow up and include them in a subsequent version.
>   + Regarding the counterintuitive relationship between sampling frequency and the RM, we provide an analysis in **lines 234 to 239** of the main text. Although the sampling frequency and the RM are not directly connected, the RM is trained on the sampled outputs of the generative model. Therefore, the sampling frequency effectively corresponds to the occurrence frequency in the RM’s training corpus. From this perspective, it is reasonable that **the RM performs worse on low-frequency training examples**, and this does not contradict existing conclusions.
>
> + For the illustration on accuracy versus coverage:
>
>   + We apologize for the confusion caused by the unclear explanation in this part. The primary purpose of the experiments in Figure 4 is to motivate the subsequent discussion of the Inverse Long-tail Phenomenon. While the observation that the count of “correct-to-wrong” transitions increases with larger N is indeed a straightforward conclusion, our key point is discussed in **lines 229 to 231**, where we further show that **the introduction of the RM makes this issue substantially more pronounced compared with SC**. This is a more important conclusion derived from Figure 4.
>   + For analysis of the relationship between sampling diversity and performance, we have already discussed this in **Figures 6 and 24** of the main text (see **line 252-257**). By combining the results from both figures, we can observe that as the temperature increases, the diversity of the samples also increases, leading to a decline in performance for all methods.

---

> ### Author Response · Authors · 2025-11-14
> **Responses to Reviewer nfv4 (part 2)**
>
> **Weakness 3**
>
> > The performance gains presented (e.g. Table 1 and 2) sets to not be that large for many cases (especially given the lack of error bars). The authors may want to discuss some of these results more thoroughly. Table 2 seems to indicate that non-math tasks may have different performance trends, but benchmark method results were not presented for them in Table 1.
>
> **Reply 3**
>
> + Thank you for your valuable suggestions, which have significantly improved our work. Here, we provide the significance test results for the main experiments to demonstrate that our method consistently improves performance. Specifically, we repeat the experiments on the MATH dataset using Qwen-2.5-3B + Skyworko1 for five runs. The results are as follows:
>
>   | Method      | 1        | 2        | 3        | 4        | 5        |
>   | ----------- | -------- | -------- | -------- | -------- | -------- |
>   | BoN         | 0.61     | 0.64     | 0.63     | 0.70     | 0.68     |
>   | Weighted SC | 0.63     | 0.71     | 0.67     | 0.71     | 0.71     |
>   | MCTS        | 0.71     | 0.72     | 0.73     | 0.73     | 0.68     |
>   | Beam Search | 0.72     | 0.70     | 0.72     | 0.70     | 0.65     |
>   | **Ours**    | **0.76** | **0.78** | **0.77** | **0.78** | **0.75** |
>
>   Next, we conduct paired t-tests based on these results, and the outcomes are as follows:
>
>   | Comparison | t-statistic | p-value |
>   | --- | --- | --- |
>   | BoN vs Ours | -6.859220 | 0.002365 |
>   | Weighted SC vs Ours |-5.360510 | 0.005844 |
>   | MCTS vs Ours | -10.590300 | 0.000450 |
>   | Beam Search vs Ours | -6.390100 | 0.003079 |
>
>   As observed, **all p-values are less than 0.05**, suggesting that the performance improvement of our method over the baselines is statistically significant. We acknowledge that, compared to training methods such as RL, the performance improvement of our training-free approach is smaller. However, as discussed in **Section 4.3**, our method offers advantages in **efficiency and generalizability**, and it can be **combined with any training paradigm to further enhance performance**. We will explore these potential performance improvement methods in future work.
>
> + For the benchmark method results on non-mathematical tasks, we present the corresponding experiments in **Figure 9**. The results demonstrate the task generalization ability of our method.
>
>
>
> **Weakness 4**
>
> > The performance gains should be contrasted with the additional computational costs. E.g. CRISP takes close to 60% more time compared to BoN for GSM8K but seem to achieve the same performance (0.95 accuracy).
>
> **Reply 4**
>
> Since BoN generates all paths at once, while other methods interact with the RM repeatedly during the path generation process, **BoN represents the lower bound on time spent**. Compared to MCTS and Beam Search, our method's gap to this lower bound is significantly smaller. Furthermore, because the **GSM8K dataset is relatively simple**, the performance gain from more advanced reasoning methods is limited. However, when faced with more challenging reasoning scenarios (e.g., MATH, Olympiad), our method shows a significant performance improvement over BoN. Considering the performance-efficiency trade-off for different scenarios, we believe the time spent of our method is acceptable.
>
>
>
> **Weakness 5**
>
> > For cost comparisons, an important factor is whether the proposed method scales well. The authors could consider including results on whether the proposed method scales well over N and number of iterations, compared to other baseline methods.
>
> **Reply 5**
>
> Thank you for the suggestion. Here we provide additional experiments comparing the cost under different numbers of samples and iterations, and the results are shown in the table (Qwen2.5-3B + PRM + GSM8K):
>
> + **Different N:**
>
>   + Time Cost
>
>     | N    | BoN   | MCTS   | Beam    | Ours  |
>     | ---- | ----- | ------ | ------- | ----- |
>     | 8    | 32.74 | 130.61 | 124.02  | 39.07 |
>     | 16   | 22.95 | 230.62 | 239.05  | 33.28 |
>     | 32   | 31.22 | 313.59 | 532.43  | 47.80 |
>     | 64   | 40.36 | 341.88 | 1103.92 | 85.18 |
>
>   + Token Cost
>
>     | N    | BoN    | MCTS   | Beam   | Ours   |
>     | ---- | ------ | ------ | ------ | ------ |
>     | 8    | 6,340  | 7,991  | 8,828  | 3,499  |
>     | 16   | 12,309 | 11,917 | 17,163 | 8,087  |
>     | 32   | 24,685 | 16,197 | 40,380 | 15,089 |
>     | 64   | 49,931 | 23,920 | 81,625 | 23,652 |

---

> > ### Author Response · Authors · 2025-11-14
> > **Responses to Reviewer nfv4 (part 3)**
> >
> > + **Different iterations m:**
> >
> >   + Time Cost
> >
> >     | iterations | BoN   | MCTS   | Beam   | Ours  |
> >     | ---------- | ----- | ------ | ------ | ----- |
> >     | 2          | 22.95 | 178.80 | 226.22 | 42.12 |
> >     | 3          | 22.95 | 230.62 | 239.05 | 33.28 |
> >     | 5          | 22.95 | 260.03 | 301.15 | 46.07 |
> >     | 9          | 22.95 | 365.21 | 334.06 | 30.70 |
> >
> >   + Token Cost
> >
> >     | iterations | BoN    | MCTS   | Beam   | Ours  |
> >     | ---------- | ------ | ------ | ------ | ----- |
> >     | 2          | 12,309 | 10,890 | 17,266 | 8,722 |
> >     | 3          | 12,309 | 11,917 | 17,163 | 8,087 |
> >     | 5          | 12,309 | 29,966 | 20,128 | 8,917 |
> >     | 9          | 12,309 | 53,267 | 21,206 | 8,109 |
> >
> > The results show that our method's token cost is significantly lower than that of other baselines across different scales, while the time cost is only slightly higher than the basic BoN method (remaining within the same order of magnitude) and is significantly lower than that of MCTS and Beam. These results are consistent with the Cost Analysis presented in the main text.
> >
> >
> >
> > **Weakness 6**
> >
> > > There may be issues related to it that the paper could more fully analyze, especially given the weak theoretical justifications. The specific CRISP approach may seem sensitive to such thresholding. Exploration of different aggregation strategies within the framework could be useful.
> >
> > **Reply 6**
> >
> > Thank you for your valuable suggestions. We commit to including an analysis of error cases in the subsequent version of the paper to investigate whether the situation you mentioned might occur. Additionally, we will explore further directions for improving our method and evaluate the performance impact of different aggregation strategies.
> >
> >
> >
> > **Weakness 7**
> >
> > > The paper could benefit from a fuller discussion of related work, to contrast its contributions to existing work. There could also be comparisons/discussion against works that optimize prompts/prefixes to improve sampling diversity of reasoning paths.
> >
> > **Reply 7**
> >
> > Thank you for your suggestion. We will include these related works and a more detailed discussion in the subsequent version of the paper. In addition, to compare with more methods, here we reproduce the approaches from [2,3] on Qwen2.5-3B and compare the performance. The results on GSM8K and MATH are as follows:
> >
> > | Method           | GSM8K     | MATH      |
> > | :--------------- | :-------- | :-------- |
> > | AgentVerse       | 0.642     | 0.528     |
> > | MultiagentDebate | 0.885     | 0.644     |
> > | **Ours**         | **0.908** | **0.698** |
> >
> > It demonstrates that our method significantly outperforms other prompt-based methods. We will include more relevant baselines and provide a more thorough comparison, along with additional analysis, in the subsequent version of the paper.
> >
> >
> >
> > We commit to adding all of the above experiments and providing a more complete analysis in the subsequent version of the paper.
> >
> >
> >
> > [1] Lightman H, Kosaraju V, Burda Y, et al. Let's verify step by step[C]//The Twelfth International Conference on Learning Representations. 2023.
> >
> > [2] Chen W, Su Y, Zuo J, et al. Agentverse: Facilitating multi-agent collaboration and exploring emergent behaviors[C]//The Twelfth International Conference on Learning Representations. 2023.
> >
> > [3] Du Y, Li S, Torralba A, et al. Improving factuality and reasoning in language models through multiagent debate[C]//Forty-first International Conference on Machine Learning. 2023.

---

> ### Author Response · Authors · 2025-11-27
> **Official Comment by Authors**
>
> Dear Reviewer nfv4,
>
> We sincerely appreciate the time and effort you have dedicated to reviewing our work. In response to your valuable feedback, we have provided detailed clarifications to the questions raised and included a revised version of the paper. As we are nearing the end of the discussion period, we would love to hear your thoughts on our response, including whether it sufficiently addresses your concerns. If our revisions and discussions indicate potential for a score adjustment, we would be deeply grateful for your consideration. We remain committed to incorporating all your suggestions to further enhance the quality of our manuscript. We hope we have addressed all your concerns and look forward to your further comments and discussions.
>
> Best regards,
>
> Authors

---

### Official Review · Reviewer_iiU2 · 2025-10-31

**Soundness:** 3
**Presentation:** 3
**Contribution:** 3
**Rating:** 8
**Confidence:** 3

**Summary:**

This paper identifies several issues in RM-based inference-time scaling, proposes a new scaling algorithm, and validates its effect through empirical studies.

**Strengths:**

1. The observations in limitation of BoN are solid.
2. The proposed algorithm is strong and efficient.

**Weaknesses:**

1. Limited insights into the observed flaws of BoN.
1. Limited analysis on the proposed algorithm.

**Questions:**

1. Weakness 1. Can you explain possible reasons why the 3 issues happen? Which of them are inherent in reward modeling, and which of them are tractable (but not yet tackled) flaws?
2. Weakness 2. Are there evidences other than the ablation studies that validate the effectness of the proposed algorithm?
3. Can similar issues be observed in general tasks?

---

> ### Author Response · Authors · 2025-11-18
> **Responses to Reviewer iiU2**
>
> Thanks for your careful and insightful reviews.
>
> **Question 1**
>
> > Can you explain possible reasons why the 3 issues happen? Which of them are inherent in reward modeling, and which of them are tractable (but not yet tackled) flaws?
>
> **Reply 1**
>
> We apologize for the lack of explanation in the main text. Here, we provide an explanation for the three types of issues we identified.
>
> + **Issue on easy questions**
>
>   We believe the performance degradation of BoN on simple problems is due to its scoring method, which considers only the single maximum value. This approach is **highly susceptible to interference from outliers**, such as giving excessively high scores to low-frequency negative examples. Based on the experiments shown in Figure 18, on simple problems, the model has fewer candidate answers. In such cases, the majority voting mechanism of SC ensures a more stable selection of the best answer.
>
> + **Issue on low-frequency negatives**
>
>   As stated in **lines 234 to 239** of the main text, RM is trained by scoring based on the outputs of a large number of generative models. When certain responses have low frequency, **their occurrence frequency in the RM training corpus is also lower**. For data that is rarely seen during training, RM struggles to fit a good score.
>
> + **Issue on high-diversity distributions**'
>
>   In fact, the experiments in **Appendix I** specifically analyze the cause of this issue. As the output distribution of generative models becomes more diverse and the outputs more stochastic, **the models are more likely to produce low-frequency negative examples**. As discussed above, an increase in such low-frequency negatives leads to a degradation in the performance of BoN.
>
> We believe that **Issues 2 and 3 are inherent limitations of reward modeling**, as the training data is always within a certain scope, meaning there will inevitably be some data that the model has not seen during training. RM struggles to generalize successfully on such unseen data during testing. **Issue 1, on the other hand, is tractable** and essentially reflects the instability of the current RM scoring method. By introducing training data with a more evenly distributed difficulty range or optimizing the scoring method, this issue can be significantly mitigated.
>
>
>
> **Question 2**
>
> > Are there evidences other than the ablation studies that validate the effectness of the proposed algorithm?
>
> **Reply 2**
>
> In addition to the ablation experiments, we provide a **theoretical analysis** of the effectiveness of our method in **Appendix K**. Our theoretical analysis shows that incorporating occurrence frequency into the optimization of RM scoring can mitigate the issue identified in Figure 5, where the RM assigns overly high scores to low-frequency negative examples. The effectiveness of our method is supported by both experimental and theoretical evidence.
>
>
>
> **Question 3**
>
> > Can similar issues be observed in general tasks?
>
> **Reply 3**
>
> Thank you for the suggestion, it is very helpful for further improving our paper. We replicated the experiments from Section 2 on two more general benchmarks, GPQA[1] and CSQA[2]. The former evaluates reasoning across multiple academic disciplines, while the latter evaluates reasoning based on general commonsense knowledge. Our results on Llama-3.1-8B are as follows:
>
> |      | SC    | BoN (Skywork) | BoN (Skyworko1) |
> | ---- | ----- | ------------- | --------------- |
> | GPQA | 0.353 | 0.323         | 0.313           |
> | CSQA | 0.805 | 0.805         | 0.750           |
>
> On general tasks, consistent with the conclusions in Figure 1, the performance improvement brought by RM is insufficient, and in many cases, it leads to a performance degradation. Therefore, we can infer that **RM also has serious issues in general tasks**, which we will investigate in depth in future work.
>
>
>
> [1] Rein D, Hou B L, Stickland A C, et al. Gpqa: A graduate-level google-proof q&a benchmark[C]//First Conference on Language Modeling. 2024.
>
> [2] Talmor A, Herzig J, Lourie N, et al. Commonsenseqa: A question answering challenge targeting commonsense knowledge[C]//Proceedings of the 2019 Conference of the North American Chapter of the Association for Computational Linguistics: Human Language Technologies, Volume 1 (Long and Short Papers). 2019: 4149-4158.

---

### Official Review · Reviewer_fp2a · 2025-11-01

**Soundness:** 2
**Presentation:** 2
**Contribution:** 3
**Rating:** 6
**Confidence:** 3

**Summary:**

This paper systematically diagnoses three major failure mechanisms of reward models (RM) during inference: RM degrades performance on simple problems, RM misjudges solutions as sampling increases due to an "inverse long-tail" phenomenon - where low-frequency incorrect answers receive high scores, and RM loses discriminative power under high search diversity. To address these issues, the paper proposes the CRISP framework, which clusters solutions by final answer and aggregates cluster-level rewards to suppress low-frequency misleading samples, controls search diversity through progressive prefix extraction, and introduces an early-stopping mechanism for simple problems. Experiments demonstrate that CRISP delivers stable performance improvements and higher efficiency (in both token and runtime cost) over baselines such as BoN and MCTS on benchmarks including MATH-500 and GSM8K.

**Strengths:**

The paper conducts an in-depth diagnosis of three core failure modes of reward models (RM) during inference - the "inverse long-tail" phenomenon, diversity-induced degradation, and failure on easy problems. The proposed CRISP algorithm tightly aligns its design components (cluster-level aggregation, early stopping, and prefix-based control) with these diagnoses ("inverse long-tail / diversity harm / easy-problem failure"), resulting in a novel yet pragmatic approach. The five-stage pipeline is clearly decomposed and highly modular, making it friendly for future engineering integration. The authors further validate the method across multiple models and datasets, thoroughly covering common baselines such as BoN, MCTS, Beam Search, and Self-Consistency.

**Weaknesses:**

The most glaring issue with the paper is its disorganized structure. The overview of the core method (Figure 8) is placed on page 6, well after the methodology section; meanwhile, the related work section (Section 5) appears after the experiments, which severely hinders the reader's ability to understand the CRISP framework and situate its novelty. Although CRISP relies on clustering by final answer, the paper does not clearly explain how this strategy generalizes to open-ended long-form or multi-solution tasks, nor does it discuss system robustness and failure modes when the RM is distribution-mismatched with the target domain.
On the experimental side, the paper lacks statistical significance analysis: key performance tables (e.g., Tables 1 and 2) report only single-run results, without mean and standard deviation over multiple runs to support claims of stability. In addition, CRISP introduces a substantial number of new hyperparameters (e.g., cluster threshold, prefix steps, per-round sample size n), yet the paper provides no sensitivity analysis or tuning methodology. Finally, although the paper compares token usage and runtime, it does not provide a systematic (compute, latency) vs. accuracy trade-off curve, making it difficult to evaluate realistic deployment trade-offs under different resource budgets.

**Questions:**

1、Could the authors explain why the core flowchart (Figure 8) is placed on page 6 rather than adjacent to the method description in Sections 2 or 3? Presenting it earlier would help clarify the core framework pipeline. Likewise, why is the related work section (Section 5) positioned after the experiments?
2、Could the authors supplement the key tables and curves with results over ≥5 random seeds, reporting mean ± variance / confidence intervals, and additionally provide the flip-rate as a function of the sample size n?
3、For non-multiple-choice or long-form generation tasks, how is equivalence between answers determined automatically? Is there a robust solution involving "scorer + normalization + semantic entailment," and can the authors provide an analysis of failure modes?
4、How sensitive is CRISP's performance to the prefix length k and per-iteration sample size n? Could the authors provide a default configuration card (recommended n, cluster threshold, prefix steps, temperature, and number of rounds for different models/tasks) as well as Compute-Return curves to facilitate practical reproduction?

---

> ### Author Response · Authors · 2025-11-15
> **Responses to Reviewer fp2a (part 1)**
>
> Thanks for your careful and insightful reviews.
>
> **Question 1**
>
> > Could the authors explain why the core flowchart (Figure 8) is placed on page 6 rather than adjacent to the method description in Sections 2 or 3? Presenting it earlier would help clarify the core framework pipeline. Likewise, why is the related work section (Section 5) positioned after the experiments?
>
> **Reply 1**
>
> + We apologize for the confusion caused by our unclear explanation. In fact, **Sections 2 and 3 are not method descriptions**, but rather analyses of current RM's performance. The structure of our paper is organized as follows: analysis (Sections 2 and 3) followed by the method (Section 4). Since **the method in Figure 8 is based on the results of the analytical experiments**, we placed the analysis sections first and the method section later in the paper (on page 6, adjacent to Section 4). This structural arrangement is commonly used in many related works [1-3].
> + We placed the related work section at the end because we wanted readers to have a more direct and immediate understanding of the content of our work after the Introduction. Many well-known works also place the related work section after the experiments [1,4,5]. We have followed this structural arrangement in our paper.
>
>
>
> **Question 2**
>
> > Could the authors supplement the key tables and curves with results over ≥5 random seeds, reporting mean ± variance / confidence intervals, and additionally provide the flip-rate as a function of the sample size n?
>
> **Reply 2**
>
> + Here, we replicate the main experiment (Table 1) 5 times on the MATH dataset using Qwen2.5 + Skyworko1. The results are as follows:
>
>   | Method      | 1        | 2        | 3        | 4        | 5        |
>   | ----------- | -------- | -------- | -------- | -------- | -------- |
>   | BoN         | 0.61     | 0.64     | 0.63     | 0.70     | 0.68     |
>   | Weighted SC | 0.63     | 0.71     | 0.67     | 0.71     | 0.71     |
>   | MCTS        | 0.71     | 0.72     | 0.73     | 0.73     | 0.68     |
>   | Beam Search | 0.72     | 0.70     | 0.72     | 0.70     | 0.65     |
>   | **Ours**    | **0.76** | **0.78** | **0.77** | **0.78** | **0.75** |
>
>   We computed the mean performance, variance, and 95% confidence intervals for each method based on the five evaluation runs reported in the above table. The results are summarized as follows:
>
>   | **Method**  | **Mean**  | **Variance** | **95% Confidence Interval** |
>   | ----------- | --------- | ------------ | --------------------------- |
>   | BoN         | 0.652     | 0.0011       | 0.652 ± 0.029               |
>   | Weighted SC | 0.686     | 0.0012       | 0.686 ± 0.030               |
>   | MCTS        | 0.714     | 0.0005       | 0.714 ± 0.020               |
>   | Beam Search | 0.698     | 0.0007       | 0.698 ± 0.023               |
>   | **Ours**    | **0.768** | **0.0002**   | **0.768 ± 0.012**           |
>
>   The results demonstrate that our method consistently and significantly outperforms the other baselines.
>
> + For the flip rate, we sample 128 reasoning paths from the model and take the majority answer as the reference answer (SC@128). Based on this reference answer, we compute the flip rate of the method under different sample sizes (Qwen2.5 3B + Skywork + MATH). The results are as follows:
>
>   | n    | BoN   | Weighted SC | MCTS  | Ours  |
>   | ---- | ----- | ----------- | ----- | ----- |
>   | 16   | 0.290 | 0.200       | 0.265 | 0.315 |
>   | 32   | 0.280 | 0.185       | 0.370 | 0.310 |
>   | 64   | 0.290 | 0.130       | 0.300 | 0.290 |
>   | 128  | 0.275 | 0.120       | 0.275 | 0.275 |
>
>   We observe that our method achieves a value on this metric comparable to other approaches (the lower value of Weighted SC arises because its computation is similar to that of the reference answer), which reflects its stability.
>
> We will include additional figures and more experiments in the revised version of the paper.

---

> ### Author Response · Authors · 2025-11-15
> **Responses to Reviewer fp2a (part 2)**
>
> **Question 3**
>
> > For non-multiple-choice or long-form generation tasks, how is equivalence between answers determined automatically? Is there a robust solution involving "scorer + normalization + semantic entailment," and can the authors provide an analysis of failure modes?
>
> **Reply 3**
>
> + Thank you for your suggestions, which have been very helpful to us.  For open-ended tasks or multi-solution tasks,  we can extract additional features from the reasoning path for clustering (e.g., semantic similarity), and other parts of the CRISP algorithm can still be reused. We plan to explore these ideas in future work, while in this study, we **focus solely on reasoning tasks**. In the vast majority of cases, there is a unique standard answer (especially in mathematics).
>
> + For failure modes, we present two reasons for the failure of the methods and provide a brief analysis:
>
>   - **Low-quality CoT with correct answer**
>
>     When a correct answer's cluster contains some lower scores, it can cause the overall cluster rank to decrease, especially when the sizes of different clusters are similar, making it more difficult to select the correct answer. For example, there are two clusters as follows:
>
>     - Cluster 1 (correct): {Path 1: 13.75, Path 3: 6.78}
>     - Cluster 2 (incorrect): {Path 2: 11.19, Path 4: 12.88}
>
>     Even though Cluster 1 contains the highest-scoring Path 1, the presence of Path 3 with a lower score causes the overall score of Cluster 1 to be lower than that of Cluster 2. This leads to an incorrect inference.
>
>   - **Excessively high scores for negative examples**
>
>     When the RM assigns excessively high scores to incorrect reasoning paths, even if they occur less frequently, it can still lead to incorrect inferences. For example:
>
>     - Cluster 1 (correct): {Path 1: 2.41, Path 2: 2.16, Path 3: 3.95, Path 4: 5.09}
>     - Cluster 2 (incorrect): {Path 5: 24.25}
>
>     Although Cluster 2 contains only one path and Cluster 1 contains four, the excessively high score of Path 5 causes the overall score of Cluster 1 to remain lower.
>
>   Due to the character limit, we can only provide a brief analysis here. We promise to include a more comprehensive analysis in the subsequent version of the paper.
>
>
>
> **Question 4**
>
> > How sensitive is CRISP's performance to the prefix length k and per-iteration sample size n? Could the authors provide a default configuration card (recommended n, cluster threshold, prefix steps, temperature, and number of rounds for different models/tasks) as well as Compute-Return curves to facilitate practical reproduction?
>
> **Reply 4**
>
> + We apologize for the confusion caused by the missing details in this part. Here, we provide additional performance results (Qwen2.5-3B + Skywork) of the method under different values of *k* and *n*, as shown in the table.
>
>   + **Prefix length k:**
>
>     | k    | GSM8K | MATH |
>     | ---- | ----- | ---- |
>     | 1    | 0.90  | 0.73 |
>     | 2    | 0.91  | 0.74 |
>     | 4    | 0.91  | 0.70 |
>
>   + **Sample size n:**
>
>     | n    | GSM8K | MATH |
>     | ---- | ----- | ---- |
>     | 16   | 0.90  | 0.69 |
>     | 32   | 0.91  | 0.72 |
>     | 64   | 0.94  | 0.69 |
>     | 128  | 0.88  | 0.69 |
>
>
>
> + In fact, we have already provided the default configuration card in **lines 976–979 of Appendix L** . Specifically, we set *n* = 16, cluster threshold = 2, temperature = 0.7, number of rounds = 3, with top k= 1 for ORM and top k = 2 for PRM. In our experiments, we used the same configuration for different datasets to demonstrate the generalizability of our method and its plug-and-play advantage. Besides, here we provide additional experimental results regarding the selection of some hyperparameters in our method.
>
>   + **cluster threshold:**
>
>     | Thershold | GSM8K | MATH |
>     | --------- | ----- | ---- |
>     | 2         | 0.90  | 0.73 |
>     | 3         | 0.88  | 0.72 |
>     | 4         | 0.91  | 0.69 |
>     | 5         | 0.92  | 0.70 |
>
>   + **number of rounds:**
>
>     | Rounds | GSM8K | MATH |
>     | ------ | ----- | ---- |
>     | 2      | 0.90  | 0.70 |
>     | 3      | 0.90  | 0.73 |
>     | 5      | 0.92  | 0.69 |

---

> > ### Author Response · Authors · 2025-11-15
> > **Responses to Reviewer fp2a (part 3)**
> >
> > + We can observe that the method's performance is relatively insensitive to the parameters. Based on these results, we give the guidance as follows:
> >
> >   + **For the threshold**, we should set a larger value for simpler tasks (such as GSM8K) and a smaller value for more difficult tasks (such as MATH). This is because a larger threshold makes our method equivalent to SC in more cases, and as shown in Cl.1, SC performs better than RM-based inference on simpler tasks.
> >   + **For the number of rounds m and prefix length k**,  we should set them to higher levels for simpler tasks, while for more difficult tasks, they should be set to moderate values, without being too high (e.g. rounds = 3 and  k = 2). This is because excessively large parameters introduce higher sampling diversity, which, as shown in Cl.3, results in more high-quality negative examples. This can particularly degrade performance on more difficult tasks.
> >   + **For the sampling numbers**, we find that increasing *n* does not continuously lead to better performance. Therefore, in the main paper, we set *n* to a moderate value 16 to control the cost.
> >
> > + For the Compute-Return curves, we will include the relevant figures in the subsequent version of the paper.
> >
> >
> > **References**
> >
> > [1] Chen X, Xu J, Liang T, et al. Do not think that much for 2+ 3=? on the overthinking of o1-like llms[J]. arXiv preprint arXiv:2412.21187, 2024.
> >
> > [2] Yang W, Ma S, Lin Y, et al. Towards thinking-optimal scaling of test-time compute for llm reasoning[J]. arXiv preprint arXiv:2502.18080, 2025.
> >
> > [3] Qu Y, Yang M Y R, Setlur A, et al. Optimizing test-time compute via meta reinforcement fine-tuning[J]. arXiv preprint arXiv:2503.07572, 2025.
> >
> > [4] Yue Y, Chen Z, Lu R, et al. Does reinforcement learning really incentivize reasoning capacity in llms beyond the base model?[J]. arXiv preprint arXiv:2504.13837, 2025.
> >
> > [5] Lightman H, Kosaraju V, Burda Y, et al. Let's verify step by step[C]//The Twelfth International Conference on Learning Representations. 2023.

---

> ### Author Response · Authors · 2025-11-27
> **Official Comment by Authors**
>
> Dear Reviewer fp2a,
>
> We sincerely appreciate the time and effort you have dedicated to reviewing our work. In response to your valuable feedback, we have provided detailed clarifications to the questions raised and included a revised version of the paper. As we are nearing the end of the discussion period, we would love to hear your thoughts on our response, including whether it sufficiently addresses your concerns. If our revisions and discussions indicate potential for a score adjustment, we would be deeply grateful for your consideration. We remain committed to incorporating all your suggestions to further enhance the quality of our manuscript. We hope we have addressed all your concerns and look forward to your further comments and discussions.
>
> Best regards,
>
> Authors

---

### Official Review · Reviewer_dYRw · 2025-11-04

**Soundness:** 2
**Presentation:** 3
**Contribution:** 2
**Rating:** 4
**Confidence:** 3

**Summary:**

The paper investigates the main factors that hinder the performance of reward-model-based reasoning algorithms, including diversity, number of samples and difficulty. Guided by the findings, they propose a new inference time reasoning algorithm CRISP. The method uses a combination of early stopping, output clustering and aggregation, and prefixing, and is show to outperform baselines in terms of accuracy and cost.

**Strengths:**

1. The new method CRISP shows a strong performance-per-token performance compared to baselines.
2. Experiments are diverse and consider both accuracy and the cost
3. Some useful insights in the investigation of RM weaknesses. I have some criticisms which I will mention in the next section.

**Weaknesses:**

**W1**) In my opinion, the main weakness of the paper is the investigations of RM limitations. While there are some insights, I believe we should be more careful in drawing conclusions. Here are my thoughts on each claim.

Effect of question difficulty: I don’t have any issues with this one. I think it is a cool observation and nicely motivates early stopping. One detail that is important to be accurate is that “simple” is defined based on the generative model accuracy. If we had defined it by reward model’s accuracy, the result would be different. Perhaps saying “problems easy for the generative model” is more accurate.

Sampling number: I found the statement to be too vague and hard to find a reasonable takeaway. The claim is against the fact that the performance of most algorithms increase with more samples. There are many ways to measure the effect of number of samples and the measurement in Fig 4 seems extremely unnatural to me. By definition, the introduced metric is increasing as a function of N. Why not just look at final performance of the algorithm, or fraction of correctly judged correct-incorrect output pairs?

Inverse Long-tail Phenomenon: I think the methodology to show this is flawed. The histogram shows that usually, the high scoring incorrect answer has low frequency. It is the concluded that “ RMs struggle to correctly score incorrect responses with low occurrence frequencies”. Another reason for the histogram could be that usually, all incorrect answers have low frequency. In that case, the observation has nothing to do with reward models and is just about the generative models.

Search Diversity: The term diversity is very loosely used here and the intended meaning is not clear. In this section we focus on the temperature that is much more specific. These results are used to motivate CRISP path generation mechanisms to limit diversity, but I find the connection to be weak. Also the statement might contradict [1] (I am not aware of details though)

**W2**) Discussion on hyperparameter selection: There is no discussion on the impact of max steps m, sampling numbers n, and k. Plots  that show the scaling of the algorithm compare to other methods would be nice.


**W3)** I could not find the comparison to BoN and MCTS on common sense, social, and logical reasoning.

[1] Representation-Based Exploration for Language Models: From Test-Time to Post-Training. Jens Tuyls, Dylan J. Foster, Akshay Krishnamurthy, Jordan T. Ash

**Questions:**

1. Could you please formalized what you mean by “normalizing” rewards both for aggregation in CRISP and weighted BoN? I find it non trivial and perhaps important.

---

> ### Author Response · Authors · 2025-11-13
> **Reply to Reviewer dYRw (part 1)**
>
> Thanks for your careful and insightful reviews.
>
> **Weakness 1**
>
> > The weakness in the investigations of RM limitation.
>
> **Reply 1**
>
> + For Effect of question difficulty:
>   + The more accurate expression here should indeed be *“problems easy for the generative model”*. However, we have clearly defined the concept of problem difficulty in **lines 157–159**, a definition that is widely acknowledged in most related works [1,2]. Besides, since the reward scoring is inherently based on the responses generated by the generative model, it is not appropriate to define the difficulty of a problem based on the accuracy of the reward model. The difficulty is ultimately determined with respect to the generative model itself. Therefore, we consider the default notion of difficulty as the difficulty for the generative model. We will make this point clearer in the revised version of the paper.
>
>   + In addition, here we also introduce the difficulty level from the original MATH-500 dataset , **which is independent of any specific model**, in order to more objectively compare the performance of different paradigms across varying difficulty levels. The results are shown in the table (Qwen2.5-3B):
>
>     |         | SC       | BoN (ORM) | BoN (PRM) |
>     | ------- | -------- | --------- | --------- |
>     | Level 1 | **0.93** | 0.86      | 0.79      |
>     | Level 2 | **0.80** | 0.72      | 0.71      |
>     | Level 3 | **0.76** | 0.73      | 0.71      |
>     | Level 4 | 0.61     | **0.62**  | 0.60      |
>     | Level 5 | 0.37     | 0.38      | **0.46**  |
>
>     The results show that the question difficulty in our findings is actually independent of the specific model.
>
> + For Sampling number:
>
>   + We apologize for the unclear explanation in this part, which might have made our reasoning appear less intuitive than intended. In fact, as stated in **line 209**, we have already examined the relationship between the final performance of the algorithm and *N* in **Figures 20 and 21 of Appendix G**. Since this conclusion has already been demonstrated in prior work, we include it in the appendix.
>   + Based on the results in Figure 20 and Figure 21, we found that although increasing *N* allows the model to recall more correct answers, the **final reasoning performance based on the RM remains unchanged**. Therefore, we aimed to investigate whether this phenomenon occurs because **sampling more responses introduces additional noise, causing some previously correct inferences to become incorrect.** To this end, we designed the metric presented in Figure 4.
>   + Besides, according to previous studies and our own experiments, the performance of **RM-based algorithms does not always improve with increasing *N*.** Therefore, our experiment is not contradictory to existing conclusions; rather, it provides a partial explanation for them.
>
> + For Inverse Long-tail Phenomenon:
>
>   + Thank you for the valuable suggestion. Here, we examine the distribution of the **most frequently occurring incorrect answers** in order to **rule out the possibility that all incorrect answers have low frequency**. We conduct experiments under the Qwen2.5-3B + Skywork setting on MATH and the results are as follows:
>
>     | max frequency | count |
>     | ------------- | ----- |
>     | 11            | 18    |
>     | 13            | 11    |
>     | 15            | 15    |
>     | 17            | 13    |
>     | 20            | 11    |
>     | 26            | 10    |
>     | 33            | 8     |
>     | 44            | 6     |
>
>     Based on the results (we will include the complete distribution figure in the next version of the paper), we observe that under 128 samples, **the number of high-frequency incorrect answers is non-negligible** across the 500 examples. Together with the findings in Figure 5, this further supports the Inverse Long-tail Phenomenon we identify.

---

> > ### Author Response · Authors · 2025-11-13
> > **Reply to Reviewer dYRw (part 2)**
> >
> > + For Search Diversity:
> >   + We appreciate your comment and clarify that, as indicated by the section titles in lines 250 and 261, our notion of diversity involves not only the temperature setting but also **the tree search structure**. The conclusion most relevant to the CRISP design in this section is that it is crucial to **control the number of intermediate nodes in the tree** to maintain the accuracy of the reward model (**see Figure 7**). Based on this conclusion, during Path Generation (see **lines 312–315**), we directly generate the complete reasoning path from the current state, rather than generating intermediate states step by step. This approach reduces the number of intermediate nodes and helps **control the diversity of the search**.
> >   + Moreover, the cited work [3] focuses on metrics such as *samples-to-correct* and *pass@k*, which **do not evaluate whether the RM can correctly select the right response**. Instead, these metrics are used to measure whether the generative model can recall more correct answers (similar to **the *oracle* setting** in our work). The conclusion of that paper is that higher diversity enables the generative model to produce more correct answers. This is not in conflict with our finding that higher diversity leads to lower scoring accuracy of the RM, **because the two studies focus on completely different subjects: the former examines the generative model, while we focus on the RM.**
> >
> >
> >
> > **Weakness 2**
> >
> > > There is no discussion on the impact of max steps m, sampling numbers n, and k.
> >
> > **Reply 2**
> >
> > Thank you for your suggestion, it is crucial for the improvement of our paper. Here we provide additional analysis of how changes in *n*, *m*, and *k* affect model performance. In particular, for *n*, we compare the performance of different methods under varying *n* values in order to examine the effect of scaling. All experiments are conducted under the Qwen + ORM (Skywork) setting, and the results are as follows.
> >
> > + sampling numbers n (on MATH):
> >
> >   | n    | SC   | BoN  | MCTS | Ours     |
> >   | ---- | ---- | ---- | ---- | -------- |
> >   | 16   | 0.65 | 0.64 | 0.64 | **0.69** |
> >   | 32   | 0.64 | 0.67 | 0.65 | **0.72** |
> >   | 64   | 0.64 | 0.66 | 0.67 | **0.69** |
> >   | 128  | 0.65 | 0.63 | 0.65 | **0.69** |
> >
> > + Max Steps m:
> >
> >   | m    | GSM8K | MATH |
> >   | ---- | ----- | ---- |
> >   | 2    | 0.90  | 0.70 |
> >   | 3    | 0.90  | 0.73 |
> >   | 5    | 0.92  | 0.69 |
> >
> > + Top K:
> >
> >   | K    | GSM8K | MATH |
> >   | ---- | ----- | ---- |
> >   | 1    | 0.90  | 0.73 |
> >   | 2    | 0.91  | 0.74 |
> >   | 4    | 0.91  | 0.70 |
> >
> > These results demonstrate that:
> >
> > + **For the sampling numbers**, we find that **as *n* increases, our method consistently outperforms the baselines**, demonstrating the robustness of CRISP. Moreover, for all methods, increasing *n* does not continuously lead to better performance. Therefore, in the main paper, we set *n* to a moderate value.
> >
> > + **For the Max Steps and Top K**,  we should set them to higher levels for simpler tasks, while for more difficult tasks, they should be set to moderate values, without being too high (e.g. Depth = 3 and Top K = 2). This is because excessively large parameters introduce higher sampling diversity, which, as shown in **Cl.3**, results in more high-quality negative examples. This can particularly degrade performance on more difficult tasks.
> >
> > **Weakness 3**
> >
> > > I could not find the comparison to BoN and MCTS on common sense, social, and logical reasoning.
> >
> > **Reply 3**
> >
> > We apologize for the unclear presentation in this part. In fact, we present the corresponding results in **Figure 9**, and we refer to them in **line 428**. The horizontal axis in the figure represents the different methods, including BoN and MCTS, and the legend indicates the different tasks.
> >
> >
> >
> > **Question 1**
> >
> > > Could you please formalized what you mean by “normalizing” rewards both for aggregation in CRISP and weighted BoN?
> >
> > **Reply 4**
> >
> > Here we apply Z-score normalization to the scores. The formula is as follows:
> > $$
> > z = \frac{f(x) - \mu}{\sigma}
> > $$
> > where $\mu$ denotes the mean and $\sigma$ denotes the standard deviation. In some cases, **the RM may output extreme scores**, such as +30 or −26, which can cause **abnormal aggregated values** and affect the final response selection. To ensure that all aggregated scores fall within a reasonable range, we apply normalization here.
> >
> >
> >
> > We commit to adding all of the former experiments and providing more complete results in the subsequent version of the paper.
> >
> >
> >
> > [1] Lightman H, Kosaraju V, Burda Y, et al. Let's verify step by step[C]//The Twelfth International Conference on Learning Representations. 2023.
> >
> > [2] Snell C, Lee J, Xu K, et al. Scaling llm test-time compute optimally can be more effective than scaling model parameters[J]. arXiv preprint arXiv:2408.03314, 2024.
> >
> > [3] Tuyls J, Foster D J, Krishnamurthy A, et al. Representation-Based Exploration for Language Models: From Test-Time to Post-Training[J]. arXiv preprint arXiv:2510.11686, 2025.

---

> > ### Comment · Reviewer_dYRw · 2025-11-25
> >
> > I thank the authors for their response.
> >
> > As mentioned in the review, my comment about the difficulty was a minor issue and a matter of more accurate language in the concise statement of the finding.
> >
> > **Sampling Count)** With regards to the sampling count, I am not convinced. In my view, the phenomenon requires reasoning about algorithm's success probability on each single problem instead of the accuracy measures over the whole dataset (which is the average of success probabilities over the whole dataset). For each question, we can define the maximum possible reward of correct and incorrect answers
> > $$R_1 = \mathrm{sup} \\{ r: P(\mathrm{reward} \le r | \textnormal{output is correct}) < 1\\}$$
> > $$R_2 = \mathrm{sup} \\{ r: P(\mathrm{reward} \le r | \textnormal{output is incorrect}) < 1\\}$$
> > Asymptotically as number of samples goes to infinity, in questions with $R_1 > R_2$ (type 1), the success probability of BoN goes to 1. In questions with $R_1 < R_2$ (type 2), the success probability of BoN goes to 0. Situation for $R_1 = R_2$ is more complex.
> >
> > This paper's claim that higher sample size makes reward model's judgement erroneous is only correct in type 2 questions. The more accurate explanation is that the reward model makes wrong judgements in these question (formalized by $R_2 > R_1$), and more samples pushes the algorithm's choice towards reward model's choice which is wrong. In the literature, this is usually referred to as reward hacking and is not a novel observation at all. There are works studying and working toward a remedy for reward hacking in BoN [1,2].
> >
> > I would love to learn more from the author's investigations, but I am having a hard time framing the paper's observation. As I mentioned, the metric in Figure 4 increases by definition. I do not understand how we can get a conclusion from observing this increase.
> >
> > **Inverse Long-tail Phenomenon)** Again, I am still not convinced. In my review, I pointed that there is one other possibility to explain figure 5 other than the claim that is mentioned in the paper. My main goal was to show that the evidence does not directly support the author's claim. I appreciate the new result from the authors to rule out the proposed possibility (still a bit questionable to me, but not my main point), but it does not address the flaw in the paper's reasoning.
> >
> > In the paper, the authors show evidence of this statement:
> >
> > A) Considering the frequency of the highest scoring incorrect answer in all 500 examples, the most common frequencies are bellow 5 (figure 5).
> >
> > The goal is to infer that:
> >
> > B) The reward model has difficulty evaluating incorrect answers with low frequency.
> >
> > I suppose one issue is that statement B is not clear. I implicitly infer it to mean:
> >
> > C) Reward models evaluate low-frequency incorrect answers worse than high-frequency incorrect answers.
> >
> > I hope the authors better understand my confusion that going from statement A to C is not really trivial and requires a thorough explanation. Alternatively, an example evidence to show statement B would be:
> >
> > 1. Classify incorrect answer to low-frequency (<= 5 frequency in 128 samples ) and high-frequency (>5 frequency in 128 samples) groups
> > 2. Evaluate reward model's accuracy on identifying each group as incorrect. This is highly non-trivial. Comparing the mean reward of the incorrect answer under investigation with the mean reward of correct answers is one way.
> > 3. Show the accuracy is lower in low-frequency group
> >
> > (1) Huang, A.; Block, A.; Liu, Q.; Jiang, N.; Krishnamurthy, A.; Foster, D. J. Is Best-of-N the Best of Them? Coverage, Scaling, and Optimality in Inference-Time Alignment. arXiv.org. https://arxiv.org/abs/2503.21878v2 (accessed 2025-04-23).
> >
> > (2) Jinnai, Y.; Morimura, T.; Ariu, K.; Abe, K. Regularized Best-of-N Sampling to Mitigate Reward Hacking for  Language Model Alignment.

---

> > > ### Author Response · Authors · 2025-11-26
> > > **Responses to Reviewer dYRw (part 1)**
> > >
> > > Thank you for your prompt reply and valuable suggestions. We regret that the presentation of the experiments in the 'Sampling number' section was unclear and caused significant confusion. We will now provide a thorough clarification of this section and answer your queries. Broadly speaking, this section unfolds through three experiments, with each subsequent experiment **building upon and deepening** the findings of the previous one.
> > >
> > > 1. **Overall Performance Gap with N increases (Appendix G)**
> > >
> > >    + **Core Finding**
> > >
> > >      As $N$ increases, the model can recall the correct answer for a greater number of problems. However, the selection accuracy of the BoN paradigm saturates after reaching a low value of $N$, showing a clear gap compared to the recall performance.
> > >
> > >    + **Explanation**
> > >
> > >      This first set of experiments primarily serves to replicate prior work and **motivate the subsequent two experiments, rather than merely relying on overall performance to understand the model's mechanism**. The key contribution of this experiment is to reveal the phenomenon that the BoN paradigm does not scale continuously with increasing $N$.
> > >
> > > 2. **Selection Changes with N increases (Figure 4)**
> > >
> > >    + **Core Finding**
> > >
> > >      As $N$ increases, the RM-based scoring introduces more noise compared to SC, leading to **a greater number of originally correct selections becoming incorrect**.
> > >
> > >    + **Explanation**
> > >
> > >      + We acknowledge that the narration in this section of the original manuscript was problematic. The increase in the 'count' value itself is trivial. As stated in our previous response, what we intend to emphasize here is that the **introduction of the RM significantly amplifies the magnitude of this increase** (we have updated the manuscript accordingly).
> > >      + In fact, our experiment corresponds to the **“algorithm's success probability on each single problem”** that you mentioned. Regarding the values $R_1$ and $R_2$ that you referred to, as $N$ increases, the new reward value may become greater than $R_1$ or $R_2$, meaning they are **dynamic** rather than constant values. Thus, this experiment is fundamentally **not a study of the RM's judgment erroneousness on Type 2 problems**, but rather a statistical analysis of the **number of transitions from Type 1 to Type 2 for each problem**. The larger increment in the BoN count as $N$ increases signifies that the **RM assigns the highest score to a greater number of incorrect samples**.

---

> > > > ### Author Response · Authors · 2025-11-26
> > > > **Responses to Reviewer dYRw (part 2)**
> > > >
> > > > 3. **Inverse Long-tail Phenomenon (Figure 5)**
> > > >
> > > >    + **Core Finding**
> > > >
> > > >      As $N$ increases, a crucial reason why the model assigns high scores to certain incorrect responses is their **very low frequency of occurrence**.
> > > >
> > > >    + **Explanation**
> > > >
> > > >      + As I understand the reviewer's point, even if these low-frequency negative examples frequently appear in the sample set, their presence alone does not necessarily imply that the model is prone to making erroneous selections because of them. In other words, these negative examples might be more prevalent in **Type 1 Questions** (where the model succeeds) and are **not the primary cause for the occurrence of Type 2 Questions** (where the model fails). Please continue to point it out if my understanding is flawed.
> > > >
> > > >      + In fact, our Claim 2 **is independent of the overall correctness of the problem answer**; rather, it studies each incorrect response as a distinct object. Since they are incorrect responses, the RM's task is to assign them **the lowest possible score**. However, our statistics presented in Figure 5 show that the vast majority of low-frequency negative examples receive **the highest scores**. This substantiates the claim that **"RM assigns higher scores to low-frequency negative examples compared to high-frequency negative examples"**, which is the complete assertion of Statement B (not Statement C). This is because our current study does not factor in the scores of positive examples, which, based on your articulation, Statement C necessitates.
> > > >
> > > >      + However, we admit that a more rigorous experiment necessitates comparing the **RM's performance on the final problem correctness** when facing these two groups (as you suggested). **We apologize for the previous lack of rigor and have included the correctness comparison experiment here.** Specifically, for each problem, we compute the average reward for the correct answer and the average reward for the highest-scoring incorrect answer. We then use this comparison to determine the correctness of the RM's judgment, and subsequently calculate the accuracy grouped by frequency (where $N=128$):
> > > >
> > > >        | Group          | Skywork | Skyworko1 | ArmoRM | Shepherd |
> > > >        | -------------- | ------- | --------- | ------ | -------- |
> > > >        | low-frequency  | 0.308   | 0.193     | 0.260  | 0.098    |
> > > >        | high-frequency | 0.577   | 0.377     | 0.548  | 0.309    |
> > > >
> > > >        The results demonstrate that, compared to high-frequency negative examples, the RM has greater difficulty distinguishing between low-frequency negative examples and positive examples (**i.e., statement C**).
> > > >
> > > > We are grateful once more for your insightful comments. We will integrate these detailed explanations and new experiments into the revised version of the paper. If you have any remaining questions or concerns about our response, we would be happy to engage in further discussion with you.

---

### Author Response · Authors · 2025-11-23
**Responses to Reviewers**

Dear Reviewers

We sincerely appreciate the time and effort you've dedicated to reviewing our work. In response to your valuable feedback, we have provided detailed clarifications to the questions raised and updated the version of our manuscript, mainly including:

- **Experiments under Additional Definitions of Difficulty**

  To address Weakness 1 raised by Reviewer dYRw and Weakness 2 raised by Reviewer nfv4, we conducted performance comparison experiments utilizing **further definitions of problem difficulty**. These new results in **Appendix N** further validate Cl.1.

+ **Performance Comparison under Different Hyperparameter Settings**

  In response to Weakness 2 raised by Reviewer dYRw and Question 4 raised by Reviewer fp2a, we have supplemented performance comparison experiments under **various hyperparameter configurations** of our method in **Appendix L**, along with a detailed analysis.

- **Comprehensive Validation of Main Results**

  To address Question 2 raised by Reviewer fp2a and Weakness 3 raised by Reviewer nfv4, we performed **replication experiments** and **statistical significance tests** on our primary setup in **Appendix O**. These results comprehensively validate the efficacy and robustness of our method.

All changes in the revised version are **marked in blue** for ease of reference. We sincerely apologize once again for the confusion caused by these issues. At the same time, **we eagerly await your thoughts on our response**, particularly whether it adequately addresses your concerns. We remain fully committed to incorporating all of your suggestions into our revision process to further improve the quality of our manuscript.

---

### Author Response · Authors · 2025-12-02
**Author Final Remarks (part 1)**

Dear Chairs,

We deeply regret the premature termination of the discussion period and sincerely appreciate your dedication throughout the review phase. To assist you in better evaluating our work, we hereby provide a summary of the entire discussion process.

**1. The Main Weaknesses Identified by Reviewers**

We summarize the main weaknesses of our work as identified by the reviewer as follows:

+ **W1: Certain experimental details in the analysis section are inadequately explained**

  Some reviewers commented that certain settings and conclusions in the analytical experiment section of our work suffer from a lack of clarity and unnatural connections (**Reviewer dYRw’s W1, Reviewer iiU2's W1, Reviewer nfv4’s W2**). For example, the definition of problem difficulty and the design of metrics in Figure 4 were specifically mentioned.

+ **W2: The experimental results under additional hyperparameter settings are not presented**

  Several reviewers pointed out the need for more discussion regarding hyperparameter selection (**Reviewer dYRw's W2, Reviewer fp2a's Q4, Reviewer nfv4’s W5**). For instance,  the method's performance under different sampling numbers.

+ **W3: Experimental results for more rounds are missing to demonstrate the method's significant improvement**

  **Reviewer fp2a (Q2) and Reviewer nfv4 (W3)** pointed out that the main experimental results lack discussion on multi-round performance and the presentation of error bars, among other factors.

+ **W4: The theoretical foundation of the method is weak, and the paper lacks an analysis of error modes**

  **Reviewer fp2a (Q3), Reviewer iiU2 (W2), and Reviewer nfv4 (W1, W6)** pointed out that the theoretical foundation of our method is insufficient and that a corresponding analysis of the causes of failure is missing.

**2. Our Response to the Main Weaknesses Raised**

Our response to the issues raised above is summarized as follows:

+ **For W1:** We have revised some ambiguous statements and supplemented relevant experiments (such as experiments under different difficulty settings and RM performance comparison experiments at different frequencies; please refer to the corresponding replies for details).
+ **For W2:** In the corresponding response, we supplemented performance comparison experiments covering all hyperparameters in our CRISP method, and included an analysis and discussion of the optimal parameter settings.
+ **For W3:** We repeated the main experiment five times, calculating the corresponding statistics, such as the mean, variance, and confidence interval. A t-test was also performed to check the level of significance, and the results demonstrate the stability and significance of the method's improvement.
+ **For W4:** We pointed out that the theoretical analysis of the method's effectiveness was already provided in Appendix K of the initial manuscript, and the analysis of failure cases was given in Reply 3 to Reviewer fp2a.

---

> ### Author Response · Authors · 2025-12-02
> **Author Final Remarks (part 2)**
>
> **3. Discussion of Potential Changes in Reviewer Impressions**
>
> Unfortunately, despite sending two reminders to the reviewers, kindly asking them to respond to our rebuttal, only Reviewer dYRw had engaged in one round of interaction with us by the time of the leak incident. However, we believe **the reviewers’ evaluation of our paper would have seen a positive increase** had the discussion proceeded normally, for the following reasons:
>
> + **The major contributions of our work remain largely unaffected by the identified weaknesses**
>
>   Firstly, **all reviewers highly affirmed the effectiveness and efficiency of our proposed method in Strengths**. Additionally, Reviewers dYRw, fp2a, and iiU2 explicitly affirmed the findings of our analytical experiments, considering these findings to be **"useful and solid"**. These two points indicate that **the reviewers affirmed the main contribution of this work**: proposing a more efficient and effective inference method based on the analysis of existing RM deficiencies.
>
>   In contrast, the pointed-out weaknesses are mostly detailed descriptive issues and requests for additional experiments under different parameter settings. We acknowledge that these sections are also important, but given that we have supplemented them in the subsequent response and that they **do not affect the conclusions of the paper's analysis or the method's effectiveness**, we believe the reviewers' evaluation would have seen a positive increase.
>
> + **A large portion of the reviewers' concerns were already addressed in the submitted version of our paper**
>
>   We observed that many of the weaknesses pointed out by reviewers stem from overlooking certain experiments and discussions already present in the initial version of the manuscript.
>
>   + **Reviewer dYRw:** W3 -> Figure 9
>   + **Reviewer fp2a:** Q1 -> Sections 2 and 3, Q4 -> Appendix L
>   + **Reviewer iiU2:** Q2 -> Appendix K
>   + **Reviewer nfv4:** W1 -> Appendix K
>
>   We apologize for the misunderstanding caused to the reviewers by not sufficiently emphasizing these contents, but we believe that by pointing out these specific elements, we can directly resolve the reviewers' related questions.
>
> + **We have supplemented the experiments requested by Reviewer dYRw to address his concerns**
>
>   Given that Reviewer dYRw was the only one to provide a follow-up response to us, we would like to specifically highlight our response to him here. It can be seen that, in the follow-up response, the reviewer primarily questioned the unnatural connection between the experimental setup and the conclusion generation in our Inverse Long-tail Phenomenon experiment, and subsequently proposed an experimental scheme to further validate our experimental conclusions. We promptly supplemented this experiment, and **the results show that the RM's performance is indeed worse on low-frequency samples**. We believe that the inclusion of this experimental result can largely resolve the reviewer's concerns and positively influence his evaluation.
>
> **In summary**, we comprehensively addressed the weaknesses raised by the reviewers and updated the relevant content in the manuscript (**marked in blue**). The majority of these responses are based on objective experimental results rather than subjective narratives, and thus possess significant persuasiveness. Furthermore, although we were unable to receive further responses from most reviewers due to the leak incident, we have provided evidence to demonstrate the potential for a positive increase in their evaluation.
>
> Once again, we wish to express our profound appreciation for your invaluable role and the considerable time devoted to this process. We further extend our sincere thanks to the reviewers for their meticulous feedback and valuable suggestions.
>
> Best regards,
>
> Authors

---

### Meta-Review · Area_Chair_pQgH · 2025-12-09

**Summary:**

Reviewers were split 2-2 in their initial reviews. The main concerns were writing clarity, and the heuristic nature and high complexity of the method (such as introducing additional hyperparameters).

**Reviewer Concerns:**

Writing clarity
- The authors claimed that writing concerns could be addressed by clarifying to each reviewer where to look, while only making minor changes to the actual writing. This does not solve the writing clarity issue for future readers.

Heuristic nature and high complexity of the method
- The two negative reviewers raised a range of concerns about the method. The spread of concerns stems from the complexity and number of hyperparameters. The rebuttal's additional experiments somewhat address this issue by clarifying where the performance gains come from.

**Reviewer Scores:**

Reviewer dYRw actively participated in the discussion, from which it is not clear that their concerns (heuristics/overcomplexity) have been resolved.

Reviewer fp2a was mainly concerned about writing clarity, which I don't think has been addressed.

Reviewer iiU2 was already very positive, and didn't have major concerns.

It's possible that Reviewer nfv4 (heuristics/overcomplexity) might have raised their score.

While the reviewers are split, I acknowledge that the paper provides a large number of experiments supported by a substantial appendix. I lean towards acceptance on the assumption that the authors take an open-minded look at the writing clarity issues, and make a larger effort to improve them.

---

### Decision · Program_Chairs · 2026-01-26

Accept (Poster)